# Parameter-Efficient Prompt Tuning Makes Generalized and Calibrated Neural Text Retrievers

**Weng Lam Tam[†*], Xiao Liu[†*], Kaixuan Ji[†], Lilong Xue[†], Xingjian Zhang[†],**
**Yuxiao Dong[†], Jiahua Liu[‡], Maodi Hu[‡], Jie Tang[†]**

[†]Tsinghua University    [‡]Meituan

{rainatam9784,shawliu9}@gmail.com, jietang@tsinghua.edu.cn

## Abstract

Prompt tuning attempts to update few task-specific parameters in pre-trained models. It has achieved comparable performance to fine-tuning of the full parameter set on both language understanding and generation tasks. In this work, we study the problem of prompt tuning for neural text retrievers. We introduce parameter-efficient prompt tuning for text retrieval across in-domain, cross-domain, and cross-topic settings. Through an extensive analysis, we show that the strategy can mitigate the two issues—parameter-inefficiency and weak generalizability—faced by fine-tuning based retrieval methods. Notably, it can significantly improve the out-of-domain zero-shot generalization of the retrieval models. By updating only 0.1% of the model parameters, the prompt tuning strategy can help retrieval models achieve better generalization performance than traditional methods in which all parameters are updated. Finally, to facilitate research on retrievers' cross-topic generalizability, we curate and release an academic retrieval dataset with 18K query-results pairs in 87 topics, making it the largest topic-specific one to date. [1]

## 1 Introduction

Seeking for relevant texts has been a fundamental problem for a broad range of natural language processing (NLP) applications such as open-domain question answering (Chen et al., 2017), retrieval-augmented language modeling (Guu et al., 2020), and fact verification (Thorne et al., 2018). Its recent progress has been dominantly favored by the neural approaches (Karpukhin et al., 2020; Khattab and Zaharia, 2020), especially the large-scale pre-trained language models with ever-growing parameters. For example, a recent study attempts to leverage models up to 10 billion parameters (Ni

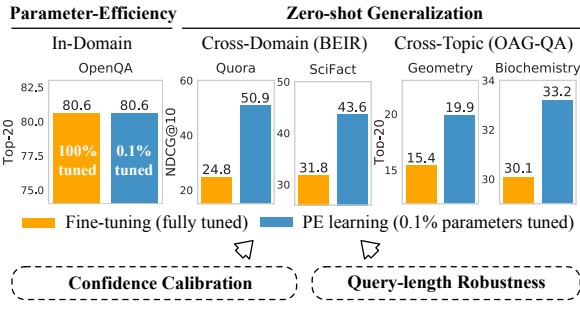

Figure 1: For DPR (Karpukhin et al., 2020) trained on OpenQA datasets, PE learning (e.g., P-Tuning v2) offers parameter-efficiency and improved generalization thanks to better calibration and query-length robustness.

et al., 2021), i.e., 100× larger than those used previously (Karpukhin et al., 2020).

Meanwhile, an increasing number of studies have focused on the **parameter-efficiency** and **generalizability** challenges of neural methods. In terms of parameter-efficiency, the common practices (Karpukhin et al., 2020) rely on fine-tuning dual encoders for queries and documents separately and thus cause parameter redundancy (Geigle et al., 2022). Furthermore, fine-tuning the full parameters of a pre-trained retriever for multi-lingual (Litschko et al., 2022) or cross-topic settings can also result in parameter-inefficiency. Moreover, despite neural approaches' in-domain outperformance, it has been found that their cross-domain generalization cannot match the simple BM25 method (Thakur et al., 2021). Consequently, these issues pose challenges to develop cost-effective neural text retrievers.

Recently, parameter-efficient (PE) transfer learning, including prompt tuning (Li and Liang, 2021; Liu et al., 2021c; Lester et al., 2021), adapters (Houlsby et al., 2019), and hybrid methods (Hu et al., 2021; Zaken et al., 2022), is proved to achieve comparable performance to fine-tuning on language understanding and generation tasks by employing very few task-specific tuning parameters. Inspired by this progress, we propose to study whether and how PE learning can benefit neural

---

[*]The first two authors contributed equally.

[1]Code and data are at https://github.com/THUDM/P-tuning-v2/tree/main/PT-Retrieval

text retrieval in terms of both parameter-efficiency and generalizability.

In this work, we systematically examine a line of mainstream PE methods in in-domain, cross-domain, and cross-topic settings. As expected, most PE approaches perform comparably to fine-tuning on in-domain retrieval. Excitingly, PE prompt tuning (Li and Liang, 2021; Liu et al., 2022) can also encourage neural text retrievers to generalize on the cross-domain benchmark BEIR (Thakur et al., 2021) and OAG-QA—a new multi-discipline academic cross-topic retrieval dataset we constructed. For example, by simply replacing fine-tuning to the parameter-efficient P-Tuning v2 (Liu et al., 2022), we achieve relative gains ranging from 3.5% to 105.0% on out-of-domain BEIR datasets.

Through empirical analyses, we attempt to provide an understanding of the better generalization brought by PE prompt tuning. First, PE prompt tuning can help empower the neural model with better confidence calibration, which refers to the theoretical principle that a model's predicted probabilities of labels should correspond to the ground-truth correctness likelihood (Guo et al., 2017). Second, it encourages better performance on queries with different lengths from in-domain training, demonstrating PE methods' generalization capacity to out-of-domain datasets.

To summarize, this work aims to advance the neural text retrievers from three aspects:

- **Problem:** we propose to leverage PE learning for neural text retrievers with much fewer tuning parameters. We demonstrate that PE prompt tuning can not only perform comparably to fine-tuning in-domain but also enable neural retrievers to achieve significant generalization advantages over fine-tuning on cross-domain and cross-topic benchmarks.

- **Understanding:** we provide an understanding of PE learning's outperformance across domains and topics. Our analysis suggests that its generalization advantage largely comes from its confidence-calibrated prediction and query-length robustness.

- **Dataset:** we construct OAG-QA, an academic paper retrieval dataset curated from real-world questions and expert answers, to test retrievers' cross-topic generalizability. With 22 disciplines and 87 topics, OAG-QA is the largest fine-grained topic retrieval dataset to date.

## 2 Related Work

**Neural Text Retrieval.** Text retrievers traditionally rely on sparse lexical-based inverted index to rank candidate documents containing query terms (e.g., TF-IDF and BM25). They benefit from the simplicity but often suffer from the lexical gap (Berger et al., 2000). Recently, neural text retrievers, including dense retrievers (Karpukhin et al., 2020; Xiong et al., 2021; Hofstätter et al., 2021), late-interaction models (Khattab and Zaharia, 2020; Santhanam et al., 2021), and hybrid or re-ranking models (Nogueira et al., 2019; Wang et al., 2020b), becomes popular as they can capture the semantic-level query-document similarity thanks to the advance of pre-trained language models (Han et al., 2021).

**Generalization in Text Retrieval.** The weaker generalizability of neural retrievers compared to conventional lexical ones has recently arouse concerns in the community (Liu et al., 2021a,b; Chen et al., 2022), and it results in BEIR, a heterogeneous cross-domain generalization benchmark (Thakur et al., 2021). While recent works notice and employ ideas like bigger pre-trained models (Ni et al., 2021) or unsupervised pre-training on large corpus (Izacard et al., 2021) to improve scores on BEIR, few of them focus on studying better transferring strategies based on existing architectures and datasets for out-of-domain generalization.

**Parameter-Efficient (PE) Learning.** Sizes of pre-trained language models are soaring up (Brown et al., 2020), causing great challenges to traditional task transfer based on full-parameter fine-tuning. A recent focus has been on the emerged PE transfer learning, including prompt tuning (Li and Liang, 2021; Liu et al., 2021c; Lester et al., 2021), adapters (Houlsby et al., 2019), and hybrid methods (Hu et al., 2021; Zaken et al., 2022). They employ very few tuning parameters to achieve fine-tuning comparable transfer performance. Despite abundant research made on problems like language understanding (Houlsby et al., 2019; Liu et al., 2022) and generation (Li and Liang, 2021), how it will impact retrieval remains under-explored.

## 3 Challenges in Neural Text Retrieval

The neural text retriever, which leverages pre-trained language models, e.g., BERT (Devlin et al., 2019) and RoBERTa (Liu et al., 2019), as the

backbone, has significantly mitigated the lexical gap (Berger et al., 2000) in text retrieval and become a standard component for many NLP applications (Chen et al., 2017; Guu et al., 2020; Petroni et al., 2021). It consists of several different categories and in this work we focus on the following two dominant ones.

- **Dense Retriever** (Karpukhin et al., 2020): Dense retrieval learns dual encoders to map queries and documents into a dense vector space such that relevant pairs of queries and documents have shorter distances. It usually adopts the inner-dot product for the sake of efficiency as $\text{sim}(q, p) = E_Q(q)^T E_P(p)$ where $E_Q(\cdot)$ and $E_P(\cdot)$ are dense encoders that map queries and documents to dense vectors, respectively. A rule-of-thumb training objective is the Noise Contrastive Error (NCE), which takes the query $q_i$ and its relevant (positive) document $p_i^+$ and $n$ irrelevant (negative) documents $p_{i,j}^-$ as:

$$\mathcal{L}_{\text{NCE}} = -\log \frac{e^{\text{sim}(q_i, p_i^+)}}{e^{\text{sim}(q_i, p_i^+)} + \sum_{j=1}^{n} e^{\text{sim}(q_i, p_{i,j}^-)}}$$
(1)

- **Late-Interaction Retriever** (Khattab and Zaharia, 2020): ColBERT combines the strengths of the bi-encoder and cross-encoder to encode the the query and document at a finer granularity into multi-vector representations. The relevance is estimated by using the rich yet scalable interaction between the query and document representations. Specifically, the model produces an embedding for every token in queries and documents and compute the relevance using the sum of maximum similarities between vectors of query tokens and all document tokens as:

$$\text{sim}(q, p) = \sum_{i \in ||E_q||} \max_{j \in ||E_d||} E_{d_j}^T E_{q_i}$$
(2)

where $E_q$ and $E_d$ are the sequences of embeddings for query $q$ and document $d$.

**Challenges.** Neural retrieval approaches, such as dense retrievers and late-interaction models, have achieved outperformance over lexical ones on typical open-domain question answering datasets, e.g., NaturalQuestions (Kwiatkowski et al., 2019). However, recent studies (Litschko et al., 2022; Thakur et al., 2021) unveil some of their inherent limitations, posing the following challenges:

- **Parameter Inefficiency:** Though the full-parameter fine-tuning empowers neural retriev-

ers to achieve good results, it results in substantial parameter redundancy from two aspects. First, training dual-encoders double the size of the parameters to be tuned. The improving strategies, such as parameter sharing (Yan et al., 2021; Geigle et al., 2022), have to sacrifice the retrieval performance. Second, the cross-lingual (Litschko et al., 2022) and cross-domain (Thakur et al., 2021) transfer may require additional full-parameter tuning on each of the individual tasks and consequently increase the number of parameters by several times.

- **Weak Generalizability:** Though neural retrievers offers advantages on domain datasets, e.g., OpenQA datasets (Karpukhin et al., 2020), some of them—particularly dense retrievers—cannot generalize well to zero-shot cross-domain benchmarks (Thakur et al., 2021). However, the zero-shot setting is widely adopted in downstream scenarios, as constructing retrieval training datasets with annotations could be outrageously expensive. Such challenge also broadly connects to the generalizability of neural networks.

In this work, we aim to explore the solutions for addressing the above challenges in neural text retrieval. Specifically, we focus on the parameter-efficient transfer learning, which has offered alternative strategies for the downstream usage of pre-trained models in natural language processing.

## 4 Parameter-Efficient Transfer Learning

We introduce the parameter-efficient transfer learning (PE learning) framework and notable techniques. Different from fine-tuning (Devlin et al., 2019), which updates the full parameters of pre-trained models for each target task, PE learning aims to achieve comparable performance to fine-tuning by tuning only a small portion of parameters per task (Houlsby et al., 2019; Li and Liang, 2021; Liu et al., 2022).

### 4.1 Transformers

The success of PE learning largely takes advantages of the Transformer architecture (Vaswani et al., 2017). Transformers are composed of stacked layers, each containing a multi-head attention module and a feed-forward network (FFN). The attention function can be written as:

$$\text{Attention}(x) = \text{softmax}(\frac{QK^T}{\sqrt{d_k}})V$$
(3)

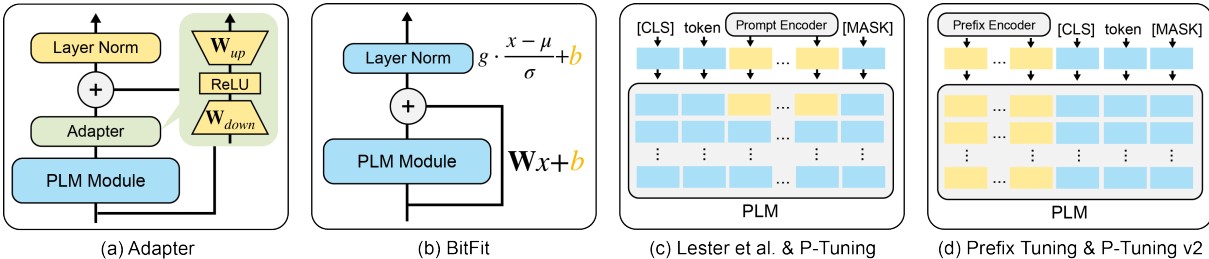

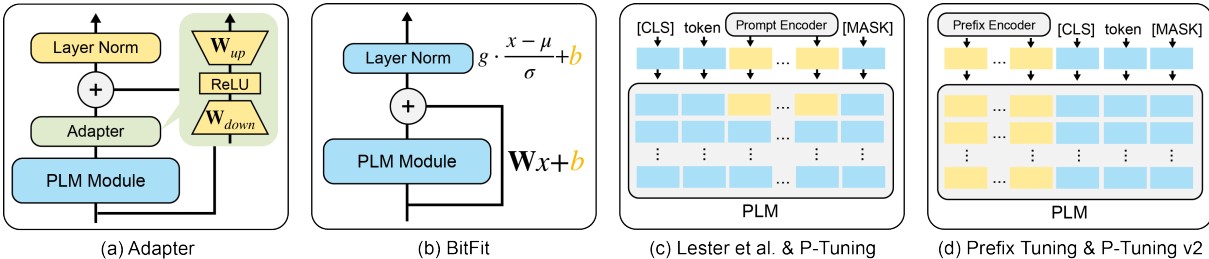

Figure 2: The illustration of four parameter-efficient methods. The PLM module represents a certain sublayer of a PLM, e.g., the attention or FFN. The components in blue are frozen and the yellow ones are trainable.

where the query $Q$, key $K$ and value $V$ are:

$$\{Q, K, V\}(x) = W_{\{q,k,v\}}x + b_{\{q,k,v\}} \quad (4)$$

The multi-head attention performs $N$ heads in parallel and concatenates their outputs to form the input to FFN where $f$ is an activation function:

$$\text{FFN}(x) = f(xW_1 + b_1)W_2 + b_2 \quad (5)$$

Different PE learning methods attempt to modify different modules of a Transformer to achieve parameter efficiency.

## 4.2 Parameter-Efficient Learning Methods

We introduce several emerging PE learning methods. Figure 2 illustrates the technical differences between them.

**Adapters (Houlsby et al., 2019; Pfeiffer et al., 2020).** The adapter inserts small modules between Transformer layers, which forms as a bottleneck to limit the amount of parameters in the format of:

$$h \leftarrow h + f(hW_{\text{down}})W_{up} \quad (6)$$

where $h$ is the input, $W_{\text{down}} \in \mathrm{R}^{d \times r}$ and $W_{up} \in \mathrm{R}^{r \times d}$ are project matrices, and $f(\cdot)$ is the activation function (Cf. Figure 2 (a)).

**BitFit (Zaken et al., 2022).** Each Transformer layer consists of self-attention, FFN, and Layer-Norm operations, all of which have certain bias terms as shown in Eqs 4 and 5. Bit-fit proposes to only tune the bias terms $b(\cdot)$ of the Transformer (Cf. Figure 2 (d)).

**Lester et al. & P-Tuning (Liu et al., 2021c).** This approach inserts trainable continuous prompts to the input sequences of the Transformer. Given a PLM, $e(\cdot)$ is the input embedding function that maps input tokens to input embeddings. For a template $T = \{[P_{0:i}], x, [P_{i+1:m}], y\}$ where $x$ is the context and $y$ is the target, e.g., the [MASK] token, the model's inputs are:

$$\{h_0, h_1, ...h_i, e(x), h_{i+1}, ..., h_m, e(y)\} \quad (7)$$

where $h_i$ is the trainable prompt (Cf. Figure 2 (b)).

**Prefix-Tuning (Li and Liang, 2021) & P-Tuning v2 (Liu et al., 2022).** Prefix-tuning concatenates $l$ trainable key and value embeddings of the attention to the prefix on each layer of the language models. Specifically, given the original key vectors $K \in \mathrm{R}^{l \times d}$ and value vectors $V \in \mathrm{R}^{l \times d}$, the trainable vectors $P_k, P_v$ are correspondingly concatenated to $K$ and $V$. The computation of an attention head becomes:

$$head_i(x) = \text{Attention}(xW^{(i)}, [P_k^{(i)} : K^{(i)}], [P_v^{(i)} : V^{(i)}]) \quad (8)$$

Here the superscript $(i)$ refers to the part of the vectors that correspond to the $i$-th head. It has been empirically proved comparable to fine-tuning on a wide range of downstream tasks, including text generation (Li and Liang, 2021), natural language understanding (NLU) and sequence labeling (Liu et al., 2022).

Since the retrieval task is more related to NLU, we employ P-Tuning v2's implementation, which makes several optimizations on top of prefix-tuning (Cf. Figure 2 (c)).

## 5 In-Domain Parameter-Efficiency

In this section, we describe the data and settings we used for the in-domain OpenQA experiments and evaluate the retrieval performance of the parameter-efficient methods introduced above.

**Datasets.** We follow (Karpukhin et al., 2020) to use five open-QA datasets and their train/test-/valid splits: Natural Questions (NQ) (Kwiatkowski et al., 2019), TriviaQA (Joshi et al., 2017), WebQuestions (WQ) (Berant et al., 2013), CuratedTREC (TREC) (Baudis and Sedivý, 2015) and SQuAD v1.1 (Rajpurkar et al., 2016). We follow (Karpukhin et al., 2020) to use the split text blocks from the English Wikipedia dump as the retrieval candidate set, which contains 21,015,324 passages.

**Settings.** We evaluate the in-domain performance

Table 1: In-domain parameter-efficiency. The retrievers are multi-task fine-tuned or PE trained on 4 OpenQA datasets (except for SQuAD*, which is excluded from Avg.) following the setting in (Karpukhin et al., 2020).

| Retrievers | #Params | Top-20 | | | | | | Top-100 | | | | | |
|---|---|---|---|---|---|---|---|---|---|---|---|---|---|
| | | Avg. | NQ | Trivia | WQ | TREC | SQuAD* | Avg. | NQ | Trivia | WQ | TREC | SQuAD* |
| BM25 | - | 63.0 | 59.1 | 66.9 | 55.0 | 70.9 | 68.8 | 76.4 | 73.7 | 76.7 | 71.1 | 84.1 | 80.0 |
| Fine-tuning | 100% | **80.6** | 79.4 | **78.8** | 75.0 | **89.1** | 51.6 | 86.9 | 86.0 | 84.7 | 82.9 | 93.9 | 67.6 |
| P-Tuning v2 | 0.1% | **80.6** | 79.5 | **78.8** | 75.2 | 88.8 | 54.6 | **87.5** | **86.6** | **85.0** | **83.3** | **95.1** | 69.6 |
| Adapter[1] | 0.8% | 38.8 | 37.1 | 38.8 | 30.3 | 49.1 | 28.3 | 56.9 | 53.8 | 56.4 | 47.5 | 70.0 | 44.1 |
| Lester et al. & P-tuning | 0.01% | 78.0 | 76.7 | 75.6 | 72.3 | 87.5 | **56.1** | 85.8 | 85.0 | 82.8 | 82.1 | 93.1 | 70.6 |
| BitFit | 0.09% | 79.9 | 78.8 | 77.6 | 74.8 | 88.2 | 56.0 | 87.2 | 86.3 | 84.5 | 83.3 | 94.5 | **71.4** |

[1] We adopt (Pfeiffer et al., 2020)'s implementation (Cf. Appendix B.1) and tried several hyper-parameter combinations.

using the Dense Passage Retrieval (DPR) Model proposed by (Karpukhin et al., 2020). We train our DPR model with four different PE learning techniques: Adapters (Houlsby et al., 2019), Lester et al. & P-Tuning (Liu et al., 2021c), P-Tuning v2 (Liu et al., 2022) and BitFit (Zaken et al., 2022), which are introduced in Section 4.2, and compare them against the original full-parameter fine-tuned DPR. Following (Karpukhin et al., 2020), we evaluate in the multi-dataset setting where the training data combines all datasets excluding SQuAD while the testing is done for all datasets.

We use top-$k$ retrieval accuracy as our evaluation metric, which measures the percentage of questions that have at least one document containing the answer in the top $k$ retrieved documents. In our experiments, we report top-20 and top-100 accuracy following (Karpukhin et al., 2020).

**Results.** We identify the best-performed hyper-parameters for each method and the results are shown in Table 1. P-Tuning v2 and BitFit are comparable to fine-tuned baseline on all datasets as expected. P-Tuning v2 also performs the best on four in-domain datasets among the tested PE approaches. On the other hand, Lester et al. & P-Tuning performs a bit weaker than the fine-tuned baseline. Adapter shows weak performance, but might be attributed to the version of implementation (Pfeiffer et al., 2020) (i.e., other versions with different implementation or more tunable parameters may be better). The results empirically demonstrate that PE methods can significantly cut down necessary tuning parameters to 0.1% and provide competitive performance in in-domain data.

Interestingly, we also notice that on the out-of-domain dataset SQuAD, P-Tuning v2, Lester et al. & P-Tuning, and BitFit substantially outperform the fine-tuned counterpart.

# 6 Cross-Domain and Cross-Topic Generalizability

In this section, we examine the zero-shot generalizability of fine-tuning and PE learning. We take P-Tuning v2 (Liu et al., 2022) as an representative for PE methods, which has the highest average in-domain accuracy . Particularly, as previous work seldom looks into the cross-topic generalization, we introduce OAG-QA, the largest fine-grained cross-topic retrieval dataset to date. On cross-domain evaluation, we adopt the well-acknowledge BEIR (Thakur et al., 2021) benchmark.

## 6.1 OAG-QA: A Fine-Grained Cross-Topic Scientific Literature Retrieval Dataset

OAG-QA is a fine-grained topic-specific passage retrieval dataset constructed by collecting high-quality questions and answers from Online Question-and-Answers (Q&A) forums, such as Quora and Stack Exchange. These forums offer people chances to ask questions and receive answers from other expert users, potentially with reference to academic papers. These references can be consequently aligned to paper entities with rich meta-information (e.g. abstract, field-of-study (FOS)) in the Open Academic Graph (OAG) (Zhang et al., 2019), the largest publicly available academic entity graph to date.

We collect questions from two influential websites: Stack Exchange[2] in English, and Zhihu[3] in Chinese. On top of the collected pairs of questions and paper titles, we align them to OAG (Zhang et al., 2019; Wang et al., 2020a; Tang et al., 2008) paper ids via public API[4]. In terms of topics, disciplines from Stack Exchange and tags from Zhihu naturally serve as fine-grained topics attached to

---

[2] https://stackexchange.com/sites
[3] https://www.zhihu.com
[4] https://www.aminer.cn/restful_service

Table 2: Examples of disciplines, topics, and example query-paper pairs (only titles are shown) in OAG-QA.

| Disc. | #Topic | Example Topic | #Query | Example query-paper pairs |
|---|---|---|---|---|
| Neural Network | 2 | Artificial Neural Network | 488 | **Q: Can neural networks be used to prove conjectures?** Paper: *Generating Correctness Proofs with Neural Networks* |
| Quantum Mechanics | 12 | Photon | 125 | **Q: What is the effective potential for photons in $X$-ray diffraction?** Paper: *Introduction to the theory of x-ray matter interaction* |

Table 3: **OAG-QA's statistics and examples.** Compared to existing scientific retrieval dataset (SciFact (Wadden et al., 2020), SCIDOCS (Cohan et al., 2020), TREC-COVID (Voorhees et al., 2021)).

| Dataset | #Query | #Corpus | #Disc. | #Topic | Fabrication |
|---|---|---|---|---|---|
| SciFact | 1,409 | 5,183 | - | - | Crowd-Source |
| SCIDOCS | 22,000 | 25,657 | - | - | User Clicks |
| TREC-COVID | 50 | 171,332 | - | - | Crowd-Source |
| OAG-QA | 17,948 | 870,000 | 22 | 87 | Online Forum |

collected questions after post-processing. For more construction details, please refer to Appendix A.

Consequently, we present OAG-QA (Cf. Table 3) which consists of 17,948 unique queries from 22 scientific disciplines and 87 fine-grained topics. Given each topic, we sample 10,000 candidate papers including the groundtruth from the same disciplines as OAG annotates, and take their titles and abstracts as the corpus.

## 6.2 Zero-Shot Cross-Domain Generalization

**Datasets.** We adopt Benchmarking-IR (BEIR) proposed in (Thakur et al., 2021), a zero-shot generalization benchmark for evaluating retrievers tasks across domains. It consists of zero-shot evaluation datasets, (15 out of 18 are available) from 9 retrieval tasks of heterogeneity. The datasets vary from each other in corpus sizes (3.6k - 15M documents), queries and documents' lengths, and domains (news articles vs. scientific papers).

**Settings.** Following (Thakur et al., 2021), we trained the models on one dataset and report the zero-shot performances on the other datasets. We choose DPR (Karpukhin et al., 2020) from dense retrievers and ColBERT (Khattab and Zaharia, 2020) from late-interaction models to explore the retrieval effectiveness under PE and full-parameter fine-tuning settings. Following the settings of BEIR, we use the open-sourced Multi-dataset DPR checkpoint (Karpukhin et al., 2020) and ColBERT model trained on MS MARCO (Nguyen et al., 2016).

To obtain comparable evaluation across datasets and tasks in BEIR (Thakur et al., 2021), we use Normalized Cumulative Discount Gain (nDCG@k)

to involve both binary and graded relevance measures for ranking quality.

**Results.** Table 4 reports the results of DPR and ColBERT on the 15 datasets of BEIR. For DPR, P-Tuning v2 generalizes much better than the fine-tuned one on all datasets except for MS MARCO and DBPedia. We observe that the datasets where our method improves by more than 5 points, such as Touche-2020 and SciFact, usually consist of long documents with average lengths over 200. We conjecture that the DPR trained on OpenQA has been biased to the 100-word document length in the oridinary setting. In summary, P-Tuning v2 achieves an absolute 5.2% improvement on the fine-tuned baseline on average. Thus, P-Tuning v2 greatly improves the out-of-domain generalization of dense retrieval models.

On the other hand, ColBERT trained by P-Tuning v2 also outperforms the fine-tuned ColBERT on almost all (13/15) datasets. P-Tuning v2 slightly underperforms on NQ and Quora where documents are relatively short. For the out-of-domain average scores, P-Tuning v2 outperforms the baseline ColBERT by an absolute gain of 2.4%. Compared to DPR, fine-tuned ColBERT generalizes better, probably because it is trained on the larger and more diverse MS MARCO and its architecture can be more scalable. But P-Tuning v2 still gains an advancement on generalization over the fine-tuned one. In conclusion, the results show that with similar in-domain performance, P-Tuning v2 can improve zero-shot generalization for cross-domain compared to fine-tuning.

## 6.3 Zero-Shot Cross-Topic Generalization

In addition to cross-domain generalization, cross-topic generalization is a more pragmatic and meaningful challenge for retrieval tasks. For example, in a scientific literature retrieval system, the corpus sizes, abstract lengths, and writing styles would not vary too much. The challenge lies in refining retrievers for more fine-grained fields-of-study.

**Settings.** We use the same trained DPR (Karpukhin et al., 2020) and ColBERT (Khattab and Zaharia,

Table 4: Zero-shot cross-domain generalization evaluated on 14 datasets of BEIR (Thakur et al., 2021). All scores are **nDCG@10**, and those of "FT" are taken from BEIR's report. ("*" denotes in-domain datasets; "FT" denotes fine-tuning; "PT2" denotes P-Tuning v2)

| Model(→) | Lexical | Dense | | Late-Interaction | |
|---|---|---|---|---|---|
| | BM25 | DPR | | ColBERT | |
| Dataset(↓) | - | FT | PT2 | FT | PT2 |
| MS MARCO | 0.228 | **0.177** | 0.171 | 0.401* | **0.414*** |
| TREC-COVID | 0.656 | 0.332 | **0.394** | 0.677 | **0.679** |
| NFCorpus | 0.325 | 0.189 | **0.224** | 0.305 | **0.327** |
| NQ | 0.329 | 0.474* | **0.479*** | **0.524** | 0.515 |
| HotpotQA | 0.603 | 0.391 | **0.416** | 0.593 | **0.623** |
| FiQA | 0.236 | 0.112 | **0.128** | 0.317 | **0.333** |
| ArguAna | 0.315 | 0.175 | **0.214** | 0.233 | **0.415** |
| Touche-2020 | 0.367 | 0.131 | **0.207** | 0.202 | **0.236** |
| CQADupStack | 0.299 | 0.153 | **0.158** | 0.350 | **0.366** |
| Quora | 0.789 | 0.248 | **0.509** | **0.854** | 0.845 |
| DBPedia | 0.313 | **0.263** | 0.254 | 0.392 | **0.407** |
| SCIDOCS | 0.158 | 0.077 | **0.099** | 0.145 | **0.156** |
| FEVER | 0.753 | 0.562 | **0.593** | 0.771 | **0.779** |
| ClimateFEVER | 0.213 | 0.148 | **0.194** | 0.184 | **0.190** |
| SciFact | 0.665 | 0.318 | **0.436** | 0.671 | **0.685** |
| Avg(w/o MS MARCO) | 0.430 | 0.255 | **0.307** | 0.444 | **0.468** |

Table 5: Zero-shot cross-topic generalization evaluated on 22 disciplines of OAG-QA. All scores are **Top-20**. ("FT" denotes fine-tuning; "PT2" denotes P-Tuning v2)

| Model(→) | Dense | | Late-Interaction | |
|---|---|---|---|---|
| | DPR | | ColBERT | |
| Topic(↓) | FT | PT2 | FT | PT2 |
| Geometry | 0.154 | **0.199** | 0.303 | **0.323** |
| Statistics | 0.149 | **0.184** | 0.289 | **0.302** |
| Algebra | **0.194** | 0.171 | **0.271** | 0.267 |
| Calculus | 0.145 | **0.169** | 0.248 | **0.259** |
| Number theory | 0.136 | **0.161** | **0.260** | 0.256 |
| Linear algebra | **0.227** | 0.211 | **0.351** | 0.345 |
| Astrophysics | 0.130 | **0.160** | 0.213 | **0.229** |
| Quantum mechanics | 0.134 | **0.169** | 0.240 | **0.245** |
| Physics | 0.205 | **0.245** | 0.349 | **0.360** |
| Chemistry | 0.157 | **0.159** | 0.296 | **0.300** |
| Biochemistry | 0.301 | **0.332** | 0.443 | **0.463** |
| Health Care | 0.367 | **0.388** | 0.446 | **0.459** |
| Natural Science | 0.306 | **0.364** | 0.408 | **0.410** |
| Psychology | 0.214 | **0.247** | 0.332 | **0.362** |
| Algorithm | 0.211 | **0.244** | 0.365 | **0.390** |
| Neural Network | 0.176 | **0.207** | 0.214 | **0.245** |
| Computer Vision | 0.152 | **0.197** | 0.264 | **0.291** |
| Data Mining | 0.139 | **0.161** | 0.226 | **0.231** |
| Deep Learning | 0.143 | **0.173** | 0.249 | **0.271** |
| Machine Learning | 0.136 | **0.187** | 0.258 | **0.278** |
| NLP | 0.149 | **0.160** | 0.234 | **0.254** |
| Economics | 0.339 | **0.353** | **0.321** | 0.298 |
| Average | 0.194 | **0.220** | 0.299 | **0.311** |

2020) model introduced in 6.2 and conduct a zero-shot evaluation. We measure top-20 retrieval accuracy on the dataset of each topic and report the average scores over each discipline.

**Results.** Table 5 compares models trained by P-Tuning v2 and fine-tuning using top-20 retrieval accuracy. P-Tuning v2 outperforms fine-tuning in 20/22 topics in DPR and 18/22 topics in ColBERT respectively. Specifically, P-Tuning v2 performs poorly in Algebra and Linear algebra, two fields which contain a large number of mathematical symbols, in both DPR and ColBERT at the same time. Overall, on average P-Tuning v2 are better than that of baseline, gaining 2.6% and 1.2% absolute improvement over DPR and ColBERT respectively.

# 7 An Understanding of the Generalization

How does PE learning help neural text retrievers to generalize well? While it might be attributed to PE learning's flatter loss minimum or alleviated catastrophic forgetting, in this work we investigate other quantifiable reasons, the *confidence calibration* and *query-length robustness*.

## 7.1 Confidence Calibration

Despite metrics like accuracy are usually the most concerned in machine learning, there are more properties to care about, such as calibration. Cal-

ibration refers to models' ability to provide class probability that corresponds to its likelihood of being true. A calibrated model provide trustful confidence to its prediction, which is particularly important for algorithms deploying in critical real-world scenarios.

Notwithstanding the higher accuracy, modern neural networks are known to be miscalibrated (Guo et al., 2017). Recent literature has also demonstrated that cross-domain calibration is a nice proxy for model's out-of-domain generalizability (Wald et al., 2021). To measure a retriever's calibration, we resort to Expected Calibration Error (ECE) proposed in (Naeini et al., 2015) as:

$$\text{ECE} = \sum_{m=1}^{M} \frac{|B_m|}{n} \left| \frac{1}{B_m} \sum_{i \in B_m} \left[ \mathbb{I}(\hat{y}_i = y_i) - \hat{p}_i \right] \right| \tag{9}$$

which bins estimates from $n$ samples within [0, 1] into $B_m$, a set of $M$ equal-length buckets. Each sample $i$ has its label $y_i$, estimated label $\hat{y}_i$, and estimated probability $\hat{p}_i$.

Following prior work (Penha and Hauff, 2021), we cast the ranking problem as multi-class classification to compute ECE. We take queries with valid top-5 predictions, apply softmax over retrieval scores per query, and turns the ranking into 5-class classification to derive ECE (Cf. Table 6) and calibration diagrams (Cf. Figure 3).

Table 6: Expected Calibration Error (ECE) (Naeini et al., 2015) of Fine-tuning (FT) and P-Tuning v2 (PT2) based on DPR (Karpukhin et al., 2020); smaller the better.

| | In-domain | | | | Cross-domain | | | | | | | | | | | | | | |
|---|---|---|---|---|---|---|---|---|---|---|---|---|---|---|---|---|---|---|---|
| | NQ | TQA | WQ | TREC | SQuAD | MS-M | TCovid | NFC | HoPo | FiQA | ArgA | T-2020 | CQA | Quora | DBPedia | SCID | FEVER | CFEVER | SciFact |
| FT | 0.135 | 0.259 | 0.219 | 0.323 | 0.114 | 0.156 | **0.164** | 0.143 | 0.071 | 0.153 | 0.156 | 0.141 | 0.144 | 0.104 | 0.128 | 0.144 | 0.069 | 0.135 | 0.120 |
| PT2 | **0.113** | **0.243** | **0.178** | **0.299** | **0.084** | **0.153** | 0.304 | **0.099** | **0.053** | **0.145** | **0.145** | **0.084** | **0.139** | **0.051** | **0.122** | **0.125** | **0.053** | **0.104** | **0.099** |

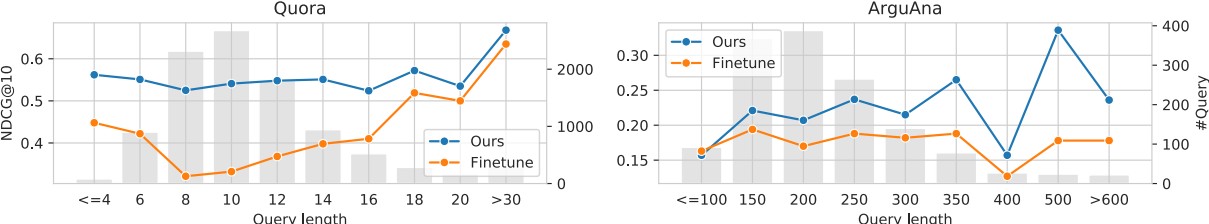

Figure 3: Calibration diagrams of DPR using P-Tuning v2 and fine-tuning on in-domain OpenQA datasets (e.g., NaturalQuestions and TriviaQA) and cross-domain BEIR datasets (e.g., ArguAna, Quora and SciFact).

Figure 4: NDCG@10 (left axis) and #Query (right axis) of P-Tuning v2 (PT2) and Fine-tuning (FT) by query length (splitted into bins) on ArguAna and Quora based on DPR (Karpukhin et al., 2020).

**Findings.** As shown in Table 6 and Figure 3, we find that P-Tuning v2 based DPR are more calibrated than its fine-tuned counterpart, whatever on in-domain or cross-domain datasets. The only exception is the TREC-COVID dataset in BEIR, which only evaluates on 50 queries and may cause a variance. To conclude, even though fine-tuning and P-Tuning v2 share a similar in-domain performance, their levels of calibration still vary largely from each other, which accords with observations in (Guo et al., 2017) that better accuracy does not mean better calibration property. Such calibration can explain P-Tuning v2's generalizability, as (Wald et al., 2021) theoretically proves that a superior multi-domain calibration effect to fine-tuning usually leads to better cross-domain generalization.

### 7.2 Query-Length Robustness

Mismatched query lengths across datasets is another hidden reason. For example, in four OpenQA datasets we experiment, most query lengths locate in the interval from 8 to 40; while other datasets can have very different query lengths. Fine-tuning changes pre-trained models parameters and may consequently bias text retrievers to certain query lengths; PE methods are free from such worries.

**Findings.** We present a case study on two typical datasets, Quora and ArguAna from BEIR (Thakur et al., 2021), to justify the hypothesis. The query lengths are derived from splitting plain query texts by white-spaces. For a clearer visualization, we split the lengths by equal-sized bins. As shown in Figure 4, when queries are medium-length (30-100), both P-Tuning v2 and fine-tuning perform comparably. But when queries are either relatively short (in Quora) or long (in ArguAna), P-Tuning v2 generalizes much better than fine-tuning. This indicates that PE learning based (e.g., P-Tuning v2) neural text retrievers have a better robustness against varied query lengths in testing.

## 8  Conclusion

We propose to leverage PE prompt tuning for neural text retrieval, which is proved for the first time in this problem for comparable performance to full-parameter fine-tuning. Furthermore, PE approaches like P-Tuning v2 improve cross-domain and cross-topic generalization, which fundamentally comes from improved confidence calibration and query length robustness as we show. Finally, we construct and release the largest fine-grained topic-specific academic retrieval dataset OAG-QA, which contains 87 different domains and 17,948 query-paper pairs, to support future research.

## Limitations

In this section we discuss several potentially unresolved topics related to this work.

First, despite the superior parameter-efficiency of PE learning, a long-standing challenge is that it converges slower and is relatively more sensitive to hyper-parameters (typically, learning rate) than fine-tuning. We have the same observation in this work and have to bypass the problem by training longer and trying multiple groups of hyper-parameters. It is thus important to design more robust and stable training strategies for prompt tuning in the future.

Second, the OAG-QA dataset requires further exploration. As indicated in Table 7, we purposely leave 20 samples in each fine-grained topic for future investigations on the effectiveness of PE learning in few-shot and meta learning settings. Conventionally, these settings require fine-tuning the whole model in each task separately, causing great redundancy. However, PE learning's extreme parameter efficiency can come to its rescue. We leave this investigation for future work.

Third, PE learning's calibration and generalization properties should ideally be applicable to other language tasks, such as text understanding and generation. In this work, we focus on neural text retrieval, as it usually faces more distribution-shift scenarios. However, many other practical problems also suffer from the challenges of biased training data and generalization, and the application of PE learning on them remains largely unexplored.

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

# A    Details of OAG-QA

In this section, we introduced the steps for building our fine-grained cross-topic dataset OAG-QA.

## A.1    Data Collecting

OAG-QA is collected from two widely used websites: Stack Exchange in English and Zhihu in Chinese. Stack Exchange consists of various forums for specific domain, such as data science, physics and chemistry, where questions are marked by fine-grained tags by users. Zhihu is not divided by domains but questions are also tagged by topics.

## A.2    Data Pre-Processing

**Paper Extraction and Title Retrieval.** We extract the paper from answers by regular expression patterns for paper URLs. So far, we focus on five types of URLs from the answer context: *arxiv.org*, *dl.acm.org*, *doi.org*, *researchgate.net*, *www.nature.com*, which can indicate publications cited by users. Then we retrieve the titles from the URLs using the the strategies listed below:

- *arxiv.org*: We recognize pdf suffix in the URL and extract arxiv id with regular expression, then query in the arxiv API with arxiv id to get the paper title.

- *dl.acm.org*: We get the HTML with URL, use the text in "title" label, then delete the website name in the suffix, take the result as the paper title.

- *doi.org*: We extract doi with regular expression, then query the doi api with the doi to get the paper title.

- *researchgate.net*: We just split the suffix of URL into words with "_" as the paper title.

- *www.nature.com*: We get the HTML with URL, use the text in "title" label, then delete the website

name in the suffix, take the result as the paper title.

**Translation.** Because questions from Zhihu are in Chinese, we use Tencent Cloud[5] for the corpus translation.

**Cleaning.** Out of consideration for remaining the diversity of questions and difficulty to evaluate the quality of questions in academic fields, we just use simple cleaning strategies. For the questions from Stack Exchange, we deleted the questions shorter than 4 words which usually not able to restrict the topic to an appropriately sized field for paper retrieval. For the questions from Zhihu, we also just removed the questions manually which are obviously not related to academic topics.

## A.3    Alignment.

We align the extracted papers with the OAG paper database (Zhang et al., 2019) to retrieve more information of papers, especially abstract. The papers which cannot be found in the database or whose corresponding abstract is missing in the database are discarded. Finally we only keep the question-paper pairs with complete title and abstract text.

## A.4    Statistics

Our self-construct dataset OAG-QA composes of 17,948 unique questions from 21 scientific discipline and 87 fine-grained topics. We sample 10,000 papers including the groundtruth papers to construct a candidate set for each topic. The queries in each topic is divided as a training set of size 20 and a test set with the remaining data. OAG-QA has a two-level hierarchical structure where each topic is under a specific discipline. Table 7 shows the statistics of OAG-QA in detail.

# B    Implementation Details

## B.1    Implementation of DPR

**Experiment enviroment** We conducted our experiments on the Linux platform, the version of which was 3.10.0-957.el7.x86_64, and the GPU version was NVIDIA Corporation GV100GL [Tesla V100 PCIe 32GB]. After installation of CUDA 11.2, we set basic experiment environment with conda 4.10.1. Our models were implemented using Python 3.8 and PyTorch 1.11.0. We used the transformers library (version 4.12.5) for the pre-trained

---

[5]https://cloud.tencent.com/document/product/551/32572

Table 7: Statistics of OAG-QA.

| Discipline | Topic | #Query | Train | Test | #Query |
|---|---|---|---|---|---|
| Geometry | geometry | 230 | 20 | 210 | |
| | algebraic_geometry | 188 | 20 | 168 | |
| | algebraic_topology | 131 | 20 | 111 | |
| | differential_geometry | 230 | 20 | 210 | 1380 |
| | group_theory | 248 | 20 | 228 | |
| | category | 191 | 20 | 171 | |
| | topology | 162 | 20 | 142 | |
| Statistics | mathematical_statistics | 144 | 20 | 124 | |
| | bayes_theorem | 134 | 20 | 114 | 516 |
| | probability_theory | 238 | 20 | 218 | |
| Algebra | algebra | 280 | 20 | 260 | 387 |
| | polynomial | 107 | 20 | 87 | |
| Calculus | calculus | 242 | 20 | 222 | |
| | partial_differential_equation | 200 | 20 | 180 | |
| | functional_analysis | 127 | 20 | 107 | 868 |
| | hilbert_space | 127 | 20 | 107 | |
| | real_analysis | 172 | 20 | 152 | |
| Number theory | number_theory | 274 | 20 | 254 | |
| | combinatorics | 221 | 20 | 201 | 899 |
| | set_theory | 179 | 20 | 159 | |
| | prime_number | 225 | 20 | 205 | |
| Linear algebra | linear_algebra | 220 | 20 | 200 | 350 |
| | matrix | 130 | 20 | 110 | |
| Astrophysics | astronomy | 108 | 20 | 88 | |
| | astrophysics | 101 | 20 | 81 | |
| | universe | 112 | 20 | 92 | |
| | cosmology | 159 | 20 | 139 | |
| | general_relativity | 191 | 20 | 171 | |
| | special_relativity | 132 | 20 | 112 | 1575 |
| | spacetime | 172 | 20 | 152 | |
| | dark_matter | 176 | 20 | 156 | |
| | black_hole | 160 | 20 | 140 | |
| | entropy | 127 | 20 | 107 | |
| | string_theory | 137 | 20 | 117 | |
| Quantum mechanics | quantum_mechanics | 467 | 20 | 447 | |
| | quantum_entanglement | 101 | 20 | 81 | |
| | quantum_field_theory | 295 | 20 | 275 | |
| | quantum_gravity | 154 | 20 | 134 | |
| | quantum_information | 190 | 20 | 170 | |
| | particle_physics | 247 | 20 | 227 | |
| | photon | 125 | 20 | 105 | 2385 |
| | supersymmetry | 245 | 20 | 225 | |
| | thermodynamics | 213 | 20 | 193 | |
| | experimental_physics | 143 | 20 | 123 | |
| | conformal_field_theory | 101 | 20 | 81 | |
| | gauge_theory | 104 | 20 | 84 | |
| Physics | classical_mechanics | 115 | 20 | 95 | |
| | condensed_matter_physics | 201 | 20 | 181 | |
| | optics | 151 | 20 | 131 | 862 |
| | electromagnetism | 224 | 20 | 204 | |
| | mathematical_physics | 171 | 20 | 151 | |
| Chemistry | organic_chemistry | 332 | 20 | 312 | |
| | chemical_synthesis | 240 | 20 | 220 | |
| | inorganic_chemistry | 218 | 20 | 198 | 1082 |
| | physical_chemistry | 190 | 20 | 170 | |
| | computational_chemistry | 102 | 20 | 82 | |
| Biochemistry | biochemistry | 129 | 20 | 109 | 442 |
| | cell_biology | 313 | 20 | 293 | |
| Health Care | health_care | 288 | 20 | 268 | |
| | endocrinology | 111 | 20 | 91 | 623 |
| | physiology | 224 | 20 | 204 | |
| Natural Science | natural_science | 193 | 20 | 173 | 664 |
| | evolutionary_biology | 471 | 20 | 451 | |
| Psycology | social_psychology | 223 | 20 | 203 | 571 |
| | cognitive_neuroscience | 348 | 20 | 328 | |
| Algorithm | algorithm | 386 | 20 | 366 | 575 |
| | graph_theory | 189 | 20 | 169 | |
| Neural Network | artificial_neural_network | 488 | 20 | 468 | 590 |
| | cognitive_science | 102 | 20 | 82 | |
| Computer Vision | computer_vision | 315 | 20 | 295 | |
| | computer_graphics_images | 68 | 20 | 48 | 661 |
| | convolutional_neural_network | 278 | 20 | 258 | |
| Data Mining | data_mining | 131 | 20 | 111 | |
| | feature_selection | 130 | 20 | 110 | |
| | cross_validation | 117 | 20 | 97 | 694 |
| | time_series | 224 | 20 | 204 | |
| | cluster_analysis | 92 | 20 | 72 | |
| Deep Learning | deep_learning | 372 | 20 | 352 | |
| | optimization_algorithm | 238 | 20 | 218 | 791 |
| | reinforcement_learning | 181 | 20 | 161 | |
| Machine Learning | machine_learning | 583 | 20 | 563 | |
| | hidden_markov_model | 112 | 20 | 92 | 1208 |
| | classifier | 269 | 20 | 249 | |
| | linear_regression | 244 | 20 | 224 | |
| NLP | natural_language_processing | 305 | 20 | 285 | 587 |
| | recurrent_neural_network | 282 | 20 | 262 | |
| Economics | economics | 238 | 20 | 218 | 238 |
| Total | - | 17948 | 1740 | 16208 | 17948 |

BERT model. When training DPR with Adapter, the adapter-transformers(version 2.2.0) was used.

**Original DPR (Karpukhin et al., 2020)** We used the open-sourced DPR checkpoint trained on multi-task data with bert-base-uncased model (sequence length: 256). The results are aligned with DPR authors' reported ones in paper.

**DPR with P-tuning v2 (Liu et al., 2022).** For P-tuning v2 training, we used a batch size of 128 and a sequence length of 256. We trained the question and passage encoders, which are based on bert-based-uncased model, for up to 40 epochs for large datasets (NQ, TriviaQA, SQuAD and Multi-dataset setting) and 100 epochs for small datasets (TREC, QA) with a learning rate of 0.01 and a prefix length of 100 using Adam, linear scheduling with 5% warm-up and dropout rate 0.1.

**DPR with Lester et al. & P-Tuning (Liu et al., 2021c).** Like P-tuning v2, we used bert-based-uncased model as basic model, however, we only applied modification to the input and set the parameters of learning rate as 0.01. We tried different prefix length such as 100, 200 to test the performance of the model.

**DPR with BitFit (Zaken et al., 2022).** In BitFit training, we use the same values of batch size, sequence length, dropout rate and learning rate as in P-tuning v2 as well as the same model, bert-based-uncased model. It took 40 epochs to train the model in the same datasets using Adam Optimizer and linear scheduling with 5% warm-up. We fixed all parameters and trained only bias parameters.

**DPR with Adapter (Houlsby et al., 2019).** In the procedure of training Adapter, we set the Adapter architectures as PfeifferConfig style, and except learning rate of 3e-5 and epochs of 50, the parameters and datasets were all same as in Bit-Fit as introduced in the above paragraph. We adopt the implementation of adapter in ADAPTER-TRANSFORMER (Pfeiffer et al., 2020).

## B.2 Implementation of ColBERT

**Original ColBERT (Khattab and Zaharia, 2020)**
In full-parameter training, We adopt the parameters offered by (Khattab and Zaharia, 2020). We trained ColBERT model with a learning rate of $3 \times 10^{-6}$ with a batch size of 32. We fix the number of embeddings per query at 32 and follows (Thakur et al., 2021) to set the number of document embeddings as 300. The embedding dimension is set as 128.

The model is trained for up to 400k iterations.

**ColBERT with P-Tuning v2 (Liu et al., 2022).**
With P-tuning v2, We trained ColBERT from the parameters of bert-based-uncased for up to 400K steps on MS MARCO dataset with a learning rate of 0.01 and a prefix length of 64. We used a batch size of 32 and fixed the number of embeddings per query at 32 and the number of embeddings per document at 300. The embedding dimension is set to be 128.