# OpenReview forum: "Parameter-Efficient Prompt Tuning Makes Generalized and Calibrated Neural Text Retrievers"
_EMNLP/2023/Conference — EMNLP 2023 Findings_

### Official Review · Reviewer_TuGb · 2023-07-28

**Soundness:** 2

**Excitement:**

2: Mediocre: This paper makes marginal contributions (vs non-contemporaneous work), so I would rather not see it in the conference.

**Paper Topic And Main Contributions:**

This paper proposed to apply parameter-efficient (PE) tuning for neutral text retrievers which involves much less tuning parameters. Authors showed that PE tuned text retriever outpeformed fine-tuned one in most of the tasks on most of the datasets. Also, authors found that the neutral text retrievers trained with PE has better generalizability which performs better on cross-domain cross-topic retrieval tasks. Besides that, authors built a new fine-grained topic-specific academic retrieval dataset.

**Questions For The Authors:**

1. Besides OAG-QA, how is the performance of FT and PT2 on other similar datasets (e.g, SCIDOCS)?
2. Why does fine-tuning could bias the text retriever but the PE one would not, may need some in-depth insights?
3. What is the value of OAG-QA compared with existing ones besides its size is larger


**Reasons To Accept:**

1. This paper introduces PE tuning to neutral text retrievers which shows a new possibility to people working on neutral text retrievers
2. This paper gives some insights on why PE tuned retrievers have better generalizability on cross-topic retrieval tasks which may interest people in that area
3. This paper proposed a new dataset which could benefit the NLP community

**Reasons To Reject:**

1. This paper lacks noval methods and instead applies existing PE learning methods and compare with existing fine tuned models. Its scope of innovation is limited;
2. Both PE and FT methods are tested only in DPR model and ColBERT model, whether user's conclusion from this paper can be generalized and served as a reference to a wider range of models/approaches is unclear.
3. While this paper claims that understanding of PE learning's outperformance is one of the key contributions, it merely conducted some experiments to show that confidence calibration and query-length play an role in the performance gap between PE learning based neural text retriever and fine-tuned one. It does not provide an in-depth explanation on why and how these factors have such a impact;
4. The dataset proposed by this paper primarily serves its experiments. Its potential value for generalization to other NLP tasks and the quality of the dataset are not clear.
5. Among all PE methods, only PT2 has shown consistent superior performance compared with fine-tuned one. This raises a crucial question: if we choose different fine-tuned methods, could they surpass PT2? If so, the claim that PE method could have comparable performance to fine-tuning may collapse

**Reproducibility:**

4: Could mostly reproduce the results, but there may be some variation because of sample variance or minor variations in their interpretation of the protocol or method.

**Reviewer Confidence:**

2: Willing to defend my evaluation, but it is fairly likely that I missed some details, didn't understand some central points, or can't be sure about the novelty of the work.

---

> ### Author Rebuttal · Authors · 2023-08-29
>
> Thanks for your review and valuable suggestions. Here are our responses:
>
> 1. On results besides OAG-QA, for example SCI-DOCS
>
> Thanks for your good suggestion. In terms of SCI-DOCS and other datasets, actually we have reported their results as they present in the BEIR benchmark in Table 4. In fact, besides OAG-QA, it is in total 14 different out-of-domain datasets that we have tested. We pick out results of SCI-DOCS, SciFact, and TREC-COVID from Table 4 as below:
>
> | Model (→)   | DPR   |       | ColBERT |       |
> |-------------|-------|-------|---------|-------|
> | Dataset (↓) | FT    | PT2   | FT      | PT2   |
> | SCI-DOCS    | 0.077 | 0.099 | 0.145   | 0.156 |
> | SciFact     | 0.318 | 0.436 | 0.671   | 0.685 |
> | TREC-COVID  | 0.332 | 0.394 | 0.677   | 0.679 |
>
> 2. On more in-depth insights on fine-tuning's bias on the text retriever
>
> Thanks for your pointing it out. Due to the stringent page limit, we did not post our further analysis and findings in the main content. But we assure you we will update them to our future version. The major findings are:
> + The bias on the positional encodings: we find it unnecessary to tune positional encodings in the fine-tuning of text encoders for retrieval for generalization. In fact, in our preliminary experiments we find that keeping positional encodings untuned would help for length generalization.
> + The change of text embeddings and MLP parameters: we find the retrieval task does not introduce much new *language knowledge* in the fine-tuning, but majorly how an LM should change its attention over tokens. As a result, the tuning of word embeddings and MLP parameters could be not that important but to introduce more biases from the training data. It is also verified in our existed experiments, where adapter's performance is much poorer (since the (Pfeiffer et al., 2020)'s default implementation of adapter is only on the MLP part).
>
> We hope these findings could provide you some in-depth insights on the problem. We will add these findings and corresponding experiments in the future version of the paper.
>
> 3. OAG-QA's contributions
>
> In addition to the size of OAG-QA, OAG-QA has two important contributions:
> + OAG-QA is categorized into a two-level hierarchical structure. It divided questions into 22 scientific disciplines and 87 fine-grained topics. OAG-QA can offer deeper insights for the evaluation of academic retrieval tasks.
> + OAG-QA is the first retrieval dataset constructed from online Question-and-Answers forums' provided reference url, such as Quora and Stack Exchange. This method makes the dataset authentic but challenging, and allows researchers to further expand OAG-QA dataset, and construct similar datasets in other domains or languages without human efforts.
>
> In terms of OAG-QA's quality, we have mentioned that we have manually cleaned the questions created from the pipeline for OAG-QA's high-quality.
>
> 4. On the coverage of tested models
>
> The selection of DPR and ColBERT as our target testing model is under thoughtful consideration. Currently the neural text retrievers are mainly divided into two categories (as we mentioned in Section 3): Dense Retrievers and Late-interaction Retrievers. DPR and ColBERT are the most representative models in these categories, and other models are mostly modified based upon them. Thus, the verification over them could serve as indicative proof of our conclusions.
>
> But to assure you the solidness of our work, we have been conducting experiments using other models such as xMoCo [1] and ColBERT v2 [2]. Our preliminary results show that our conclusion still holds, and we will update the whole evaluation results in our future version.
>
> Thanks again for your thoughtful evaluations and time commitment. If your concerns are further addressed by our responses, would you please consider raising your score to support us? We cannot thank you enough for your help.
>
> References:
>
> [1] Yang N, Wei F, Jiao B, et al. xMoCo: Cross momentum contrastive learning for open-domain question answering. EMNLP 2021.
>
> [2] Santhanam K, Khattab O, Saad-Falcon J, et al. ColBERTv2: Effective and Efficient Retrieval via Lightweight Late Interaction. NAACL 2022.

---

### Official Review · Reviewer_deUw · 2023-08-04

**Soundness:** 3

**Excitement:**

4: Strong: This paper deepens the understanding of some phenomenon or lowers the barriers to an existing research direction.

**Paper Topic And Main Contributions:**

This paper discusses the limitations of current parameter-efficient prompt tuning approaches used for neural text retrieval tasks, where machines are required to seek relevant texts based on given documents or questions. Recent studies employ large pre-trained language models, which face challenges on parameter- efficiency and generalizability. The paper examines mainstream parameter-efficient methods in in-domain, cross-domain, and cross-topic settings through intensive experiments and analyses. The results showed that the parameter-efficient prompt tuning strategy provided the models with even better generalization ability than full-parameter tuning. During this research, a cross-topic dataset is constructed to facilitate the experiments on the models’ generalization ability.


**Reasons To Accept:**

1. This paper conducted intensive experiments analyzing the generalization ability of parameter-efficient prompt tuning on large pre-trained language models.
2. This paper provides an empirical understanding which locates the source of the generalization ability of parameter-efficient tuning approaches. This could support the design of future methods.
3. This paper constructs a cross-topic dataset which facilitates the experiments on models’ generalization ability.

**Reasons To Reject:**

In sections 6.2 and 6.3, the authors evaluate the abilities of zero-shot cross-domain generalization and zero-shot cross-topic generalization respectively. These experiments are done with two models, DPR  and ColBERT. But in section 5 on the experiments of in-domain parameter efficiency, only the DPR model is used. Did the authors conduct such experiments on the ColBERT and/ or other models? Is the experiment results consistent with other models?

**Reproducibility:**

4: Could mostly reproduce the results, but there may be some variation because of sample variance or minor variations in their interpretation of the protocol or method.

**Reviewer Confidence:**

3: Pretty sure, but there's a chance I missed something. Although I have a good feel for this area in general, I did not carefully check the paper's details, e.g., the math, experimental design, or novelty.

---

> ### Author Rebuttal · Authors · 2023-08-29
>
> Thank you very much for the review and appreciating our contributions. Here are our responses:
>
> 1. In-domain experiments of ColBERT and other models
>
> Thanks for your mentioning. In fact, the in-domain results of ColBERT (which is trained on MS-MARCO) has been reported in the original paper's Table 4 (FT: 0.401, PT-2: 0.414), annotated with "*". It shows that parameter-efficient prompt tuning method PT-2 even performs a bit better than fine-tuning on ColBERT in in-domain training.
>
> For other models, originally we picked DPR and ColBERT as the representatives to cover two typical types of retrievers (Dense & Late-interaction) and demonstrated PE methods' effectiveness on them. We are also conducting experiments on other models including xMoCo [1] and ColBERT v2 [2], and preliminary results show that the conclusion still holds. Due to the short rebuttal period, we haven't completed the whole evaluation and will report them in our paper's future version.
>
> Thanks again for your thoughtful evaluations and time commitment. Hope our response could address your concerns!
>
> References:
> [1] Yang N, Wei F, Jiao B, et al. xMoCo: Cross momentum contrastive learning for open-domain question answering. EMNLP 2021.
> [2] Santhanam K, Khattab O, Saad-Falcon J, et al. ColBERTv2: Effective and Efficient Retrieval via Lightweight Late Interaction. NAACL 2022.

---

### Official Review · Reviewer_DnA1 · 2023-08-05

**Soundness:** 4

**Excitement:**

4: Strong: This paper deepens the understanding of some phenomenon or lowers the barriers to an existing research direction.

**Paper Topic And Main Contributions:**

This paper is about prompt tuning for neural text retrievers. In this paper, the authors propose to leverage PE prompt tuning for text re-trieval across in-domain, cross-domain, and cross-topic settings, which is proved for the first time in this problem for comparable performance to full-parameter fine-tuning. And it can alleviate the issues of parameter inefficiency and weak generalization faced by fine-tuning based re-retrieval methods. The main contributions of this paper are as follows:
1.	This paper proposes to leverage PE learning for neural text retrieval. The PE prompt tuning can not only perform in-domain, but also enables neural retrievers to achieve significant generalization advantages over fine-tuning on cross-domain and cross-topic benchmarks.
2.	This paper provides an understanding of PE learning's outperformance across domains and topics. Furthermore, through a large number of experimental analysis, PE approaches like P-Tuning v2 improve cross-domain and cross-topic generalization, which fundamentally comes from improved confidence calibration and query length robustness.
3.	This paper constructs and releases the largest fine-grained topic-specific academic retrieval dataset OAG-QA, which contains 87 different domains and 17,948 query-paper pairs, to support future research.


**Questions For The Authors:**

A.	In the introduction, the contributions of this paper are summarized by using number 1, 2, and 3, without the need to adopt “Problem”, “Understanding”, “and Dataset”.
B.	The paper provides extensive experimental evidence to demonstrate the effectiveness of PE learning. However, it lacks an experimental analysis of the model's parameters, I hope the author can supplement the experiment in the follow-up.
C.	In Table 1, the retrieval effect of using Adapter is much lower than that of other methods. What is the reason?



**Reasons To Accept:**

1.	This paper demonstrates a high level of creativity and innovation. It provides the first evidence that utilizing PE prompt tuning for neural text retrieval achieves comparable performance to full-parameter fine-tuning. By updating only 0.1% of the model parameters, the prompt tuning strategy can help the retrieval model achieve better generalization performance than traditional methods that update all parameters.
2.	The experiments in this paper are comprehensive and effectively validate the effectiveness of the PE prompt tuning strategy.
3.	This paper curates and releases an academic retrieval dataset with 18K query-results pairs in 87 topics, making it the largest topic-specific one to date. If accepted, this paper can assist NLP researchers in exploring the application of PE prompt tuning to address other practical issues such as biased training data and weak generalization.


**Reasons To Reject:**

1.	The lack of a definite explanation from the author regarding the low retrieval results of the Adapter in Table 1 may cause confusion for readers.
2.	This paper provides substantial experimental evidence to demonstrate the effectiveness of PE learning, but it lacks experimental analysis on the parameters of the P-Tuning v2 model.


**Reproducibility:**

4: Could mostly reproduce the results, but there may be some variation because of sample variance or minor variations in their interpretation of the protocol or method.

**Reviewer Confidence:**

2: Willing to defend my evaluation, but it is fairly likely that I missed some details, didn't understand some central points, or can't be sure about the novelty of the work.

**Typos Grammar Style And Presentation Improvements:**

The author's paper demonstrates a high level of writing proficiency, and the paper reads smoothly.

---

> ### Author Rebuttal · Authors · 2023-08-29
>
> Thanks for your insightful review and suggestions! We are happy that you like the work, and you may find our detailed responses below:
>
> 1. For the experimental analysis of the model’s parameters, we conduct experiments with different PE learning settings. The number of trainable parameters in different training settings and the corresponding retrieval results on in-domain dataset NQ and out-of-domain dataset SciFact are shown in the table below. We train p-tuning v2 models with different lengths of prefix and thus different numbers of trainable parameters. They result in similar scores in both in-domain and out-of-domain datasets and considerably outperform fine-tuned models in out-of-domain data.
>
> |                          | #Prefix |    #Param   | TREC-COVID | NFCorpus |
> |--------------------------|:-------:|:-----------:|:----------:|:--------:|
> | Fine-tune                |    -    | 109,580,544 |    0.677   |   0.318  |
> | P-Tuning v2              |    64   |  1,277,952  |    0.682   |   0.327  |
> |                          |   100   |  1,941,504  |    0.681   |   0.322  |
> |                          |   180   |  3,416,064  |    0.700   |   0.323  |
> |                          |   256   |  4,816,896  |    0.700   |   0.328  |
> |                          |   300   |  5,627,904  |    0.685   |   0.329  |
> | Lester et al. & P-Tuning |    64   |   147,456   |    0.647   |   0.309  |
>
> 2. For the low retrieval effect of using Adapter in DPR, compared to P-Tuning methods and BitFit which manipulates the attention-related parameters, (Pfeiffer et al., 2020)'s implementation of adapters (which is also the most common implementation of adapters) is only on the MLP layer. We think that the retrieval training may focus on the improvement of LM's attention patterns rather than the memory pattern (which is known to be stored in MLP's parameters). As a result, the adapter performs poorly on transferring to retrieval tasks.

---

### Meta-Review · Area_Chair_sGEg · 2023-09-19

**Recommendation:** 3

**Metareview:**

This paper provides an extensive analysis of various parameter-efficient prompt tuning methods for text retrieval across in-domain, cross-domain, and cross-topic settings and shows that such approaches can mitigate the parameter-inefficiency and weak generalizability issues. Though there is no new approach proposed in this work, the extensive analysis and insights will benefit the research in this line. The authors also contributed a new dataset for academic retrieval. The authors should address the minor concerns raised by reviewers, e.g., the impact of the number of parameters for P-Tuning V2, and clarifications on why only DPR and ColBERT models are chosen.

---

### Decision · Program_Chairs · 2023-10-07

**Decision:**

Accept-Findings

**Comment:**

This paper provides an extensive analysis of various parameter-efficient prompt tuning methods for text retrieval across in-domain, cross-domain, and cross-topic settings and shows that such approaches can mitigate the parameter-inefficiency and weak generalizability issues. Though there is no new approach proposed in this work, the extensive analysis and insights will benefit the research in this line. The authors also contributed a new dataset for academic retrieval. The authors should address the minor concerns raised by reviewers, e.g., the impact of the number of parameters for P-Tuning V2, and clarifications on why only DPR and ColBERT models are chosen.